# Subdural Lesions Linking Additional Intracranial Spaces and Chronic Subdural Hematomas: A Narrative Review with Mutual Correlation and Possible Mechanisms behind High Recurrence

**DOI:** 10.3390/diagnostics13020235

**Published:** 2023-01-08

**Authors:** Muh-Shi Lin

**Affiliations:** 1Division of Neurosurgery, Department of Surgery, Kuang Tien General Hospital, Taichung 43303, Taiwan; neurosurgery2005@yahoo.com.tw; Tel.: +886-4-2665-1900; 2Department of Biotechnology and Animal Science, College of Bioresources, National Ilan University, Yilan 26047, Taiwan; 3Department of Biotechnology, College of Medical and Health Care, Hung Kuang University, Taichung 43302, Taiwan; 4Department of Health Business Administration, College of Medical and Health Care, Hung Kuang University, Taichung 43302, Taiwan

**Keywords:** chronic subdural hematoma, subdural hygroma, external hydrocephalus, high recurrent rate

## Abstract

The purpose of this study was two-fold. The first was to investigate the pathologic mechanisms underlying the formation of subdural fluid collection, an umbrella term referring to a condition commonly seen in the clinical setting. Accumulation of the cerebrospinal fluid (CSF) in the subdural space can be referred to in this disease category, disregarding the underlying source of the subdural fluid. However, in these two clinical situations, especially after trauma or brain surgery, fluid collection from the subarachnoid space (subdural hygroma) or from the ventricle to the subarachnoid space and infusion into the subdural space (external hydrocephalus), surgical management of critical patients may adopt the strategies of burr-hole, subduroperitoneal shunt, or ventriculoperitoneal shunt, which present distinctly different thoughts. Crucially, the former can be further transformed into chronic subdural hematoma (CSDH). The second significant theme was the pathogenesis of CSDH. Once the potential dural border cell (DBC) layer is separated such as if a wound is formed, the physiological mechanisms that seem to promote wound healing will resume in the subdural space as follows: coagulation, inflammation, fibroblast proliferation, neovascularization, and fibrinolysis. These aptly correspond to several key characteristics of CSDH formation such as the presence of both coagulation and fibrinolysis signals within the clot, neomembrane formation, angiogenesis, and recurrent bleeding, which contribute to CSDH failing to coagulate and absorb easily. Such a complexity of genesis and the possibility of arising from multiple pathological patterns provide a reasonable explanation for the high recurrence rate, even after surgery. Among the various complex and clinically challenging subdural lesions, namely, CSDH (confined to the subdural space alone), subdural hygroma (linked in two spaces), and external hydrocephalus (linked in three spaces), the ability to fully understand the different pathological mechanisms of each, differentiate them clinically, and devote more interventional strategies (including anti-inflammatory, anti-angiogenic, and anti-fibrinolysis) will be important themes in the future.

## 1. Preface

Chronic subdural hematomas (CSDHs) are commonly encountered in elderly individuals aged over 70 years, as they tend to be associated with brain atrophy. As the population ages worldwide, the incidence of CSDH is on the rise at an estimated rate of 1.7 to 21 per 100,000 people per year [1].

Anatomically, of the three strata of the meninges, the layer at the junction of the dura and arachnoid membranes, commonly referred to as the subdural space, is composed of the lamina of the dural border cell (DBC) layer [2]. The layer is most commonly dissected by traumatic mechanisms to constitute the subdural space, which is not very compact in its architecture. In the context of brain atrophy, CSDH may arise from either acute traumatic hematoma or spontaneous hemorrhage due to disease factors. Consequently, CSDH develops when a chronic clot collects in the subdural space for approximately three months. Moreover, disruption of the subarachnoid barrier layer architecture by external forces leads to the accumulation of cerebrospinal fluid (CSF) in the subdural space and the formation of subdural hygroma. As such, the subdural space, the so-called area of structural incompetence, is created by all three of these modalities, followed by neovascularization as CSDH develops. Repeated bleeding into the subdural space eventually triggers neurological symptoms.

This type of old liquefied clot slowly and progressively accumulates in the subdural space in the supratentorial or submeningeal areas, affecting either the eloquent or non-eloquent areas of the brain or developing motor and cognitive deficits associated with basal ganglia lesions and brainstem compression symptoms. Disease expression patterns of CSDH tend to be implicit, vague, and unclear, and it is often difficult to target a specific neurological disease at first sight [3,4,5,6,7].

In general, CSDH occurs more often with subtle and non-specific manifestations such as headaches, minor alterations in mental status, and gait disturbances. Mental status changes may be characterized by dementia, unresponsiveness, or even a slightly blurred consciousness. The local neurological symptoms of CSDH manifest as cranial nerve palsy, hemiparesis, and unilateral sensory deficits, even mimicking spinal cord lesions such as flaccid paraplegia and quadriplegia. CSDH involving a supratentorial blood clot or posterior fossa may cause the following signs of brainstem compression: constricted or dilated pupil, nystagmus, limb flaccidity, eye deviation, and even a decreased level of consciousness. Furthermore, manifestations of CSDH may include seizures that generate electrophysiological abnormalities through their effects on the cerebral cortex.

Alternative neurological disorders are often incorrectly identified as CSDH such as stroke or symptoms of transient ischemic attack (TIA) in the form of aphasia, hemiparesis, and contralateral sensory deficits. The manifestations of CSDH may be comparable to those of Parkinson’s disease in terms of neurological compression, which exacerbates neurological chronic degeneration to develop dysfunction of the fronto-pontine pathways or basal ganglia. Furthermore, persistent inflammatory degeneration of the brain leads to the development of psychiatric symptoms. The manifestations of psychiatric symptoms in CSDH include paranoia, schizophrenic-like psychosis, manic-depressive psychosis, depression, vague personality, and intellectual changes. Due to this special property, numerous disease categories can be manifested by CSDH, as if emulated; for this reason, CSDH is recognized as “the great imitator”.

It is beyond doubt that CSDH is associated with a high postoperative recurrence rate from a clinical perspective. Studies have documented an overall postoperative recurrence rate of approximately 2–37% [8]. In several large-scale series, recurrence rates were not significantly reduced: 10.1% (1313 patients, mostly burr-hole) [9], 10.0% (291 patients, burr-hole) [10], and 14.4% from a meta-analysis study (22 studies, 5566 patients) [11]. As above-mentioned, the subdural space created by the above-mentioned three main sources subsequently exacerbates the inflammatory response, promoting angiogenesis of fragile vessels (lacking a muscle layer) [12] and the generation of neomembranes. This vicious cycle allows for the microenvironment to remain in a state of anticoagulation and fibrinolysis. Ultimately, CSDH is encapsulated by neovascularization and neomembranes, which predispose patients to recurrent micro-hemorrhages. In this article, we explore important subdural lesions, their further transformation into CSDH, and the pathogenesis of CSDH.

## 2. A Potential Area Dissected by Acquired Factors, the Subdural Space—Clinical Considerations and Pathological States Derived from It

Connective tissue meninges provide adequate protection for the central nervous system (CNS) by completely encasing the brain and spinal cord at the external periphery. From an anatomical point of view, the meninges consist of three layers that run from below the skull to the parenchymal tissue of the brain: the dura, the arachnoid, and the pia mater; the linguistic expression “mater” represents the mother in Latin [13,14].

From the underlying skull junction, the dura mater consists of periosteal, meningeal, and dural border cell layers from the exterior to the interior. The dura mater, being a hard connective tissue, is densely packed with extracellular collagen in the front two dural layers above-mentioned; however, collagen fibers are lacking in the DBC layer [15].

The second layer of the meninges, the arachnoid mater, which lies beneath the dura, constitutes the following two layers: the arachnoid barrier cell layer and the arachnoid trabeculae [16]. In the arachnoid barrier cell layer, cells are tightly united across intercellular junctions (including tight junctions, desmosomes, and intermediate junctions) [17], rendering this layer a strong barrier in the meninges. The arachnoid trabeculae comprise fibroblasts [18], which act as columns of collagen tissues linking and stretching between the arachnoid barrier cell layer and the third layer of the meninges, the pia mater.

The subarachnoid space, the region where the arachnoid trabeculae are extended between the arachnoid barrier cell layer and the pia mater, is full of CSF, allowing the brain to be suspended there, affording a cushion of brain stabilization in the event of a head impact.

When subjected to external forces as described above, the DBC layer, which is devoid of collagen fibers, tends to be readily dissected, and a subdural space is thus developed. Additionally, the bridging veins, draining the blood from the cerebral cortex into the dural sinuses when crossing the dural border cell layer, have been found to be exceptionally thin-walled when crossing the marginal cell layer; the collagen fibers of the vessels are almost circumferential, arranged in the axial direction of the vessel, thus avoiding longitudinal support [19,20]. Consequently, this part of the vessel is prone to rupture when subjected to traction, leading to an acute subdural hematoma in which the blood clot leaks into the entire dissected subdural space. As previously described, apart from cases of injury, the subdural space can arise from disruption of the arachnoid barrier cell layer, which may lead to a unidirectional flow of CSF from the subarachnoid space to aggregate in the subdural space, forming a subdural hygroma [21].

Furthermore, external hydrocephalus serves as a distinctive situation similar to subdural hygroma, but is more highly characterized. In the presence of inappropriate circulation of CSF, the accumulated CSF in the subarachnoid space is further diverted to the subdural space along certain rupture sites of the arachnoid membrane (especially the basal cisterns or lamina terminalis) [22], eventually leading to a free communication of the “ventricle-subarachnoid-subdural space”.

## 3. Subdural Fluid Collection

This umbrella term refers to the state of excessive fluid of a “non-hemorrhagic” nature located in the subdural space. McConnell noted that subdural fluid collection was broadly defined as the accumulation of CSF in the subdural space, irrespective of the underlying source contributing to the subdural fluid accumulation [23].

Accordingly, collections of fluid originating from the subdural space, from the subarachnoid space, or alternatively from the ventricular-subarachnoid origin, followed by infusion into the subdural space, are also used in the scope of this term. Examples include [24] subdural effusion, benign subdural collections, subdural hygroma, idiopathic/benign hydrocephalus, external hydrocephalus, extraventricular hydrocephalus, benign enlargement of the subarachnoid space, and hypodense CSDH (mimicking subdural fluid collection on CT imaging).

However, because of the distinct pathological mechanism of the above-mentioned subdural lesions, within this widely redefined category, CSDH (confined to the subdural space unconnected to other spaces), subdural hygroma (two spaces connected), and external hydrocephalus (three spaces connected) need to be clinically distinguished.

### 3.1. Subdural Hygroma and Its Special Case—External Hydrocephalus

Traumatic subdural hygroma was first indicated by Mayo in 1894 [25]. Insults to the dural arachnoid interface and the formation of the potential space (subdural), as previously mentioned, allow the CSF, rather than blood clots, to collect in the subdural space. The development of subdural hygroma is broadly categorized as traumatic, non-traumatic, and operation-related [3,26,27,28,29]. Examples include head trauma, spontaneous subdural hematoma (such as ruptured arachnoid cyst), brain atrophy, dehydration, and occurrence after infection in infants. Moreover, subdural hematomas may be observed following decompressive craniectomy, cranioplasty or shunt placement, and spinal drainage.

A few theories exist for the development of subdural hygroma [30,31].

(i)Arachnoidal injury

The arachnoid barrier layer is vulnerable to damage by shear stresses. The arachnoid injury, along with gradients in intracranial pressure (ICP) between the cerebral hemispheres, leads to CSF leakage and creates a ball valve effect (a one-way arachnoid flap for CSF flow), ultimately contributing to fluid retention in the subdural space.

(ii)Inactivity of the blood–brain barrier (BBB)

Injury-induced disruption of the BBB leads to an increase in capillary permeability, prompting the leakage of plasma components into the subdural space. Consequently, elevation of osmotic pressure in the compartment leads to an increase in the volume of the subdural hygroma as a result of water influx.

(iii)Neomembrane formation

In such a vicious microenvironment, an increase in protein content provokes an inflammatory response and the subsequent formation of neomembranes. These subdural neomembranes augment the transmembrane pressure gradient, inducing elevated permeability of CSF in the arachnoid membrane and a subsequent increase in the volume of the subdural hygroma.

External hydrocephalus, that is, subdural effusion with hydrocephalus, as the term suggests, consists of free-flowing communication between the ventricles and the subdural space, combined with dysfunctional circulation of the CSF (that is, impaired CSF absorption from the level of the arachnoid granules). This state of disease, external hydrocephalus, was originally documented by Dandy in 1917 [32]. In clinical practice, external hydrocephalus is commonly encountered in the context of aneurysm rupture, subarachnoid hemorrhage, major head injury, and as a consequence of neurosurgical procedures such as decompressive craniectomy or aneurysm clipping [28,33]. Trauma or rupture of an aneurysm, causing subarachnoid hemorrhage (SAH) in the subsphenoid space, increases the likelihood of hydrocephalus, while head injury or surgical procedures reinforce the tear or invasion of the arachnoid membrane (especially the basal cisterns, lamina terminalis, parachiasmatic/Lilliquist membranes), ultimately facilitating communication between the ventricles and the subdural space.

Several mechanisms have been identified in the pathogenesis of external hydrocephalus [34,35]. Dysfunction of the arachnoid villi, possibly arising from immaturity or impairment caused by injury, leads to the inability to pass the continuously produced CSF into the bloodstream, provoking external hydrocephalus. Furthermore, beyond the circulation within the ventricular system, the CSF flows along the Virchow-Robin perivascular space to the meninges and cervical lymphatics, or within the subsarcoid space surrounded by the perineural sheaths of the cranial and spinal nerves, infusing into the nasopharyngeal lymphatic system. Under the circumstances of obstructive hydrocephalus, elevated ICP may interfere with the normal lymphatic drainage of CSF along the perivascular or perineural sheaths, resulting in the accumulation of fluid in the subdural space. In addition, in situations of elevated intracranial pressure, transependymal passage of CSF may occur, which passes through the ventricular ependymal layer to the interstitial space of the brain, and even to the perivascular space and perineural lymphatic channels above-mentioned, along its route to the subarachnoid space.

### 3.2. Diverse Views on the Diagnosis and Clinical Management of the Two Distinguished Subdural Lesions [22,36]

Typical CT scan findings are not consistently observed in every case of external hydrocephalus, where CSF accumulates in the subdural space and the lateral ventricles are dilated with brightness around the ventricles, but there is rather no visible ventricular dilatation. In contrast to external hydrocephalus, where subdural fluid is diverted from the ventricular CSF, subdural hygroma retains xanthochromic fluid and typically possesses a higher protein content than the CSF. Therefore, the Hohnsfield unit values for subdural effusions tend to be higher than those for external hydrocephalus when comparing subdural fluid collections. Enhanced CT and magnetic resonance imaging revealed an enhancing capsule in the subdural hygroma, but was absent in the external hydrocephalus. Specifically, it is likely that subdural hygroma may exhibit compression of the subarachnoid space ipsilateral to the fluid collection. Correspondingly, the external hydrocephalus revealed that the ipsilateral sulci and basal cisterns remained intact and did not show deformation. Moreover, CT cisternography, which can be used to ascertain the communication between the subdural space and ventricles, may serve as an additional method for differentiating between these two pathologies.

Regardless of subdural effusion alone or in conjunction with hydrocephalus, from the clinical considerations, if there is no mass effect, surgical treatment is not required; adversely, if there is a mass effect, surgical treatment should be recommended for either of the two conditions.

In patients with subdural hygroma, apparent mass effects can be mitigated by burr-hole drainage or subdural peritoneal shunts. In the setting of sole subdural hygroma without combined CSF circulation disturbance, ventriculoperitoneal shunt (VPS) may enhance the mass effect of the subdural space [37,38]. One article advocated the use of VPS for the treatment of subdural effusion including that related to the impaired CSF circulation, which is defined as a state of external hydrocephalus [39]. Accordingly, for external hydrocephalus, a dual-stage procedure involves the use of burr-hole drainage or SPS to diminish the mass effect of subdural hygroma, followed by VPS to address the clinical issues arising from hydrocephalus. Alternatively, the VPS can be applied directly to attenuate abnormal divergence from the ventricle to the subdural space [36,40].

Before proceeding to an overview of CSDH, the pathogenesis of subdural hygroma and external hydrocephalus and clinical considerations are briefly described in Figure 1.

More specifically, external hydrocephalus may exert pressure in the subdural space from additional pressure generated by the ventricular system, unlike subdural hygroma, which restricts one-way valve bypass that is not in communication with the ventricles. As such, the subdural pressure in the external hydrocephalus, approximately >15 cmH_2_O, is significantly higher than that in hygroma [28]. For external hydrocephalus, a VPS procedure may be proposed for adult patients with subdural pressures greater than 15 cmH_2_O or for pediatric patients with pressures greater than 12 cmH_2_O [36].

## 4. The Formation of Chronic Subdural Hematoma

According to the previous description, the potential space, the so-called subdural space, must be established within the brain prior to the formation of CSDH. This usually occurs due to trauma to the head, which gives rise to an acute subdural hematoma, causing the layer of the DBC to split and create this space. Alternatively, CSDH can originate from pre-existing subdural hygroma [41]. Under subdural hygroma, the subdural space may evolve into a CSDH as a result of a series of subsequent pathological processes including inflammation, neomembrane formation, and fibrinolysis. However, there are a few situations in which patients have specific medical conditions such as bleeding tendency, and acute bleeding may be generated spontaneously in the subdural space [42], which eventually results in CSDH. A series of pathological mechanisms underlying CSDH formation are summarized in the following sections.

### 4.1. Inflammatory Response Provoked by DBC Layer Damage as an Initiator of CSDH Development

In the following states, acute SDH due to head trauma, arachnoid rupture associated with trauma/brain surgery, hemorrhage due to physical bleeding tendency, or any stress that renders the brain atrophic, for example, aging, dehydration, or excessive drainage of CSF, the dural border cells may be torn [26,27,43], thus creating a subdural space that does not exist in the natural state.

As precursor cells for the development of fibroblastic connective tissue, when dural border cells are injured or further activated by residual blood clots in the post-acute phase, the remaining cells from the DBC layer, which have been torn and isolated either on the arachnoid surface or on the dura mater, may trigger a series of inflammatory reactions in the subdural space [44].

Once the inflammatory response is initiated, cytokines such as tumor necrosis factor-α (TNF-α), interleukin-1 (IL-1), and interleukin-6 (IL-6) are secreted in large quantities in the subdural space by stimulated connective tissue cells such as fibroblasts and immune cells [45,46]. In parallel, these inflammatory cells including neutrophils, lymphocytes, macrophages, and eosinophils can be recruited to the site, further contributing to the secretion of chemokines and eventually expanding the extent of this inflammatory cascade [47].

### 4.2. Inappropriate Inflammation Promotes the Production of Neomembranes

The aforementioned inflammatory response, as a protective mechanism in humans, aims to repair tissue damage. Inflammatory reactivity arising from the rupture of the DBC layer is followed by a further repair process that is equivalent to that of wound healing. Under the general repair of damaged tissues, collagen is deposited into the damaged or even mutilated tissue structure after injury via fibrosis. Scar tissue made up of collagen eventually fills in and replaces the original tissue remnants [48,49].

Under the pathological mechanism of CSDH, however, such a reaction is improper and excessive for the repair of dural border cells, and instead promotes inappropriate proliferation of these cells. Instead of repairing the pre-existing DBC layer, the inflammatory response proceeds to fibrosis at this disrupted interface. This process consists of the proliferation of fibrous connective tissue, which involves a high level of collagen composition and collagen matrix formation, culminating in the formation of new membranes comprising the outer and inner membranes, in approximately one week and three weeks, respectively [50]. This research team found precursors constituting core extracellular matrix (ECM) proteins such as procollagen peptides and glycosaminoglycans, at high levels within the subdural fluid of patients with head injury [51], suggesting a trend toward continuously increased synthesis of ECM in the meninges following trauma to the meninges after head trauma, that is, after cleavage of the DBC layer, as previously described.

### 4.3. Inflammation-Driven Angiogenesis Leads to Microhemorrhages on the Neomembrane Surrounding CSDH

With the aim of reconditioning cleaved DBC, the inflammatory cells present in the neomembrane, especially the outer membrane, attempt to repair the tissue with an inflammatory response. Angiogenesis is particularly relevant in this context, with a large number of new blood vessels available for increased metabolic demands of recruited infiltrating cells, tissue healing, and remodeling [52,53].

Notably, these new capillaries on the CSDH membrane are either thin or lack a basement membrane and are devoid of smooth muscle cells and pericytes, leaving the wall thin and in a highly permeable state [12]. This characteristic predisposes to recurrent microhemorrhages, along with cell transport, for example, erythrocytes, leukocytes, and plasma from the vessels into the subdural hematoma cavity [20,54], leading to hematoma enlargement.

During angiogenesis in CSDHs, vascular endothelial growth factor (VEGF), derived from neutrophils, macrophages in the fluid, or vascular endothelial cells in the CSDH membrane, initiates the activation of endothelial cells in the vessels, allowing for cell proliferation, cell migration, and subsequent formation of a vascular lumen [55,56]. Furthermore, matrix metallopeptidase-9 (MMP-9), a protein hydrolase that contributes to angiogenesis, has been found to increase vascular permeability and enhance the inflammatory response [57], which may contribute to CSDH enlargement. In combination with VEGF, MMPs may impair neovascularization stability, potentially contributing to a higher bleeding risk [54].

Moreover, an angiographic study identified that neovascularization was not limited to the outer membrane but was also found in the inner membrane of the hematoma [12,58]. Neovascularization from the inner membrane may be associated with CSDH recurrence after embolization of the middle meningeal artery (EMMA), an alternative treatment [59].

### 4.4. Hyperfibrinolysis Causes CSDH to Be Less Susceptible to Coagulation, Creating a Vicious Cycle along with Microbleeding from Neovascularization

CSDH, a highly variable hematoma pattern, originates from the detachment of the DBC layer, as described previously. The subsequent series of pathological reactions, subtly appear as a process of wound healing. After repeated micro-hemorrhages from neovascularization, an initial hemostatic effect is implemented as a result of normal physiological function, followed by a sequence of inflammation, remodeling, and fibrosis, where the originally formed fibrin bridge is further removed in preparation for tissue granulation [60]. As a result, coagulation and fibrinolysis occur repeatedly in CSDH, and under such a vicious cycle, a sustained hyperfibrinolytic state eventually emerges, which provides a plausible explanation for the tendency of hematoma enlargement and recurrence after surgery.

It is worth pondering that the predominant immune cells underlying the pathogenesis of CSDH are eosinophils, and not microglia, the major innate immune cells of the CNS. Eosinophils can promote mobility throughout the CSDH by releasing plasminogen to induce excessive fibrinolysis [50]. However, like many inflammatory substances with a double-edged nature, they have a corresponding role in promoting the maturation of hematomas [61].

The above-mentioned crucial pathological mechanisms with regard to the generation of CSDH, corresponding to the wound healing process, are demonstrated in Figure 2.

## 5. The Complex Pathogenesis Previously Described Accounted for the High Degree of Recurrence after Surgery

The high recurrence rate of CSDH is a pivotal issue that needs to be explored in depth. It is worth noting that the subdural space is shaped by the tearing of the DBC layer and the covering of the inner and outer membranes, which cannot be eliminated surgically but only by passively expanding the brain after CSDH evacuation. As such, in patients with neurological deficits following surgical irrigation for CSDH, the possibility of postoperative recurrence may be heightened when the subdural space persists because of delayed obliteration or because individual bleeding properties still tend to be present. The various factors affecting postoperative recurrence can be analyzed in anatomical and physiological categories [62].

Anatomic factors are broadly referred to as those that postpone occlusion of the subdural space. This concept corresponds appropriately to the summary of observations proposed by Kyeong-Seok Lee in CSDH pathogenesis:

“Any forces to shrink the brain can be the precipitating factors, while the opposite forces to expand the brain will be the inhibiting factors [63]”.

In particular, aging brains (atrophied brains are typically inflexible) and CSDHs with bilateral and skull base involvement (subdural space over-expansion) may cause a delayed return of the brain to its pre-existing status after hematoma evacuation. Compromised brain redistension often results in shifting of the brain within the delayed occluded subdural space, which may lead to tearing and bleeding of the bridging veins [64].

With respect to general physiological conditions, patients with bleeding tendencies or pathological states of coagulopathy present a higher risk of postoperative recurrence such as liver cirrhosis, leukemia, sepsis, chronic renal failure, and disseminated intravascular coagulation [65,66,67,68]. Significant physiological factors for high recurrence include blood clots that dominantly possess the hallmark characteristics underlying CSDH pathology such as hyper-angiogenic and hyper-fibrinolytic activity within CSDHs [69]. These dominant features are reflected in CT imaging. Hematomas with higher recurrence rates of high or mixed density propensity exhibit increased vascularity in the outer membrane, enhanced neomembrane, and high values of quantitative indicators of clot such as mean hematoma density (MHD) [11,64,70,71].

## 6. Concluding Remarks and Perspectives

In clinical practice, this common category of subdural lesion, subdural fluid collection, is generally described as an accumulation of CSF in the subdural space, independent of the underlying source causing the subdural effusion. Specifically, fluid collections in the subdural space can arise from the subarachnoid space (subdural hygroma, two-space communication) or from the ventricle to the subarachnoid space, which is in turn infused into the subdural space (external hydrocephalus, three-space communication). Once neurological symptoms develop that require decompression, surgical management is thought to employ a burr-hole, and shunting with subdural-peritoneal or ventriculo-peritoneal modality, as discussed in this article, needs to be clinically considered with care. Notably, VPS may augment the mass effect in the subdural space in the context of subdural hygroma not combining with CSF circulation disturbance. The distinct pathological mechanism between the above-mentioned subdural lesions renders differential diagnosis of clinical necessity.

Despite ongoing efforts in treatment strategies, patients with CSDH undeniably have a recurrence rate as high as 30% after surgery, even after surgical or nonsurgical management including MMA embolization. The main factor is the multiple pathogeneses of CSDH (acute hematoma, transformation from subdural hygroma, or spontaneous bleeding). Another reason lies in the fact that once the potential DBC layer is separated, this subdural lesion, acting like a wound, starts to undergo physiological mechanisms (coagulation, inflammation, fibroblast proliferation, neovascularization, and fibrinolysis) similar to wound healing. Both the neovascularization within the neomembranes encapsulating the hematoma cavity and the bridging veins in the involved subdural space are fragile and prone to bleeding, which makes CSDH difficult to clot and absorb, as if it were an intracranial wound that does not heal easily. Even during surgery, it is not possible to eradicate all pathogenic elements of the lesion, and residual tissue causes a high recurrence of bleeding.

Among the complex and clinically challenging subdural lesions described above involving CSDH or those connecting additional intracranial spaces such as subdural hygroma and external hydrocephalus, the capacity to thoroughly comprehend the respective pathological mechanisms, differentiate them clinically, and develop more advanced strategies (including anti-inflammatory, antiangiogenic, and antifibrinolytic) will be of great relevance in the future, and will be urgently needed to minimize the risk of postoperative recurrence in CSDH.

## Figures and Tables

**Figure 1 diagnostics-13-00235-f001:**
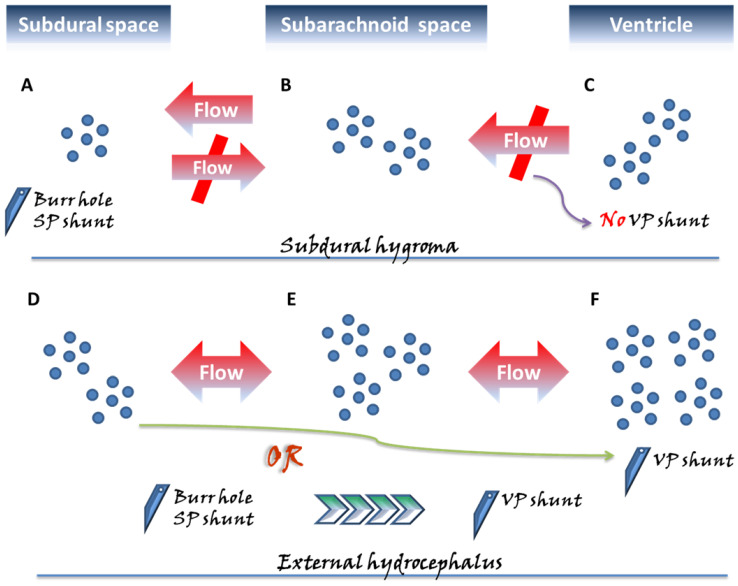
Two categories of subdural fluid collections deserving special attention, the subdural hygroma and external hydrocephalus, exhibit different pathogenic mechanisms and are surgically managed with different perspectives. In the subdural space, approximately three months following acute bleeding, a chronic subdural hematoma (CSDH) might emerge, characterized by a liquefied hematoma with neomembranes and which remains confined to the subdural space. In contrast to the above pathogenic mechanism derived from a single space, in the subdural hygroma (**A**–**C**), the flow of cerebrospinal fluid (CSF) in the subarachnoid space (**B**) further influxes unidirectionally into the subdural space (**A**) through the ruptured subarachnoid membrane. Unlike external hydrocephalus, no disturbance in CSF circulation is observed in the subdural hygroma; hence, there is no applied pressure from the ventricle (**C**). Therefore, the gradient-evoked flow of the CSF communication from (**B**) to (**A**) reaches a particular level, and then the pressures within the subdural space likely cease to increase, appearing in the one-way form, and being confined to the state of compartment pressure <15 mmHg. Moreover, in the pathological state of external hydrocephalus, more aptly denoted as subdural effusion with hydrocephalus, the pressure in the subarachnoid space (**E**) progressively builds up as a result of the disturbance in CSF caused by the hydrocephalus (**F**), driving gradient-evoked flow to gradually accumulate in this space. Afterward, in a situation similar to that of the subdural hygroma, a flow of CSF within the subarachnoid space (**E**) from the ruptured arachnoid membrane proceeds toward the subdural space (**D**). Because of the pressure caused by the hydrocephalus, the pressure gradient between the ventricle–subarachnoid–subdural space leads to a gradual gain of pressure in the subdural space. In the condition of external hydrocephalus, the pressure inside the subdural space is significantly higher, that is, approximately >15 cmH_2_O compared to subdural effusion. Based on the above-mentioned different mechanisms, in the clinical setting for patients undergoing intracranial pressure and requiring neurosurgical decompression, the use of ventriculoperitoneal (VP) shunt on subdural hygroma possibly exacerbates the compression from the subdural space due to inappropriate CSF drainage. For surgical strategies of decompression for the subdural space, burr-hole/subduroperitoneal (SP) shunt appear to be a more rationale management. Accordingly, for patients with subdural effusion with hydrocephalus, there are two aspects of the neurosurgical decompression required once the lesion has elicited neurological compromise. The surgical treatment with the VP shunt not only eliminates the hydrocephalus, but also inhibits the diversion of CSF into the subdural space. Alternatively, the two-stage surgical approach is performed with the initial burr-hole/SP shunt, and following the resolution of the neurological issues derived from the subdural space, the VP shunt is used to deal with the relevant aspects concerning the hydrocephalus.

**Figure 2 diagnostics-13-00235-f002:**
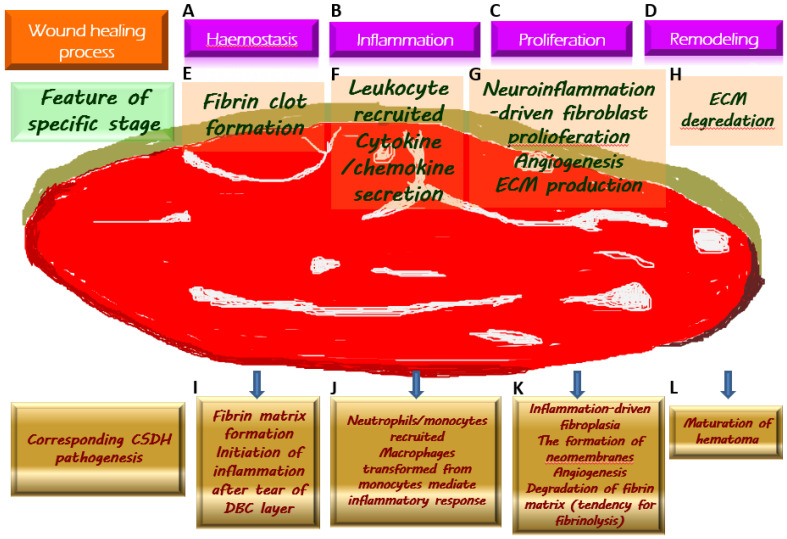
Chronic subdural hematoma (CSDH) depicted as an intracranial wound tending to not heal properly. Once a wound is created, the physiologically protective mechanism undergoes hemostasis (**A**) and the formation of fibrin clot (**E**), which serves as an extracellular matrix (ECM) for the migration of immune cells recruited by pro-inflammatory cytokines (**F**) along the subsequent inflammatory response (**B**). During the proliferative phase (**C**), fibroblasts undergo proliferation while producing ECM and generating neovascularization to provide for the nutritional needs of the cells (**G**). When the wound is about to heal, the ECM is gradually degraded by the remodeling process (**D**) and is replaced by scar tissue (**H**). Regarding the formation of CSDH, the tear of the dural border cell (DBC) layer subtly corresponds to the creation of a wound. Fibrin matrix formation provides a comprehensive explanation for the presence of coagulation signals in the effusion of CSDH (**I**). Subsequently, under the vicious cascade of inflammatory response (**J**), fibroblast proliferation, and ECM generation contribute to neomembrane production. Angiogenesis takes place in the emerging inner and outer membranes, and the newly formed vessels have fragile walls, resulting in repeated microbleeding. In addition, during this proliferative phase, the tissue that resembles granulation replaces the originally generated fibrin matrix, thereby leading to a tendency of fibrinolysis (**K**). With the maturation of the inner and outer membranes in CSDH, the neovascularization is less likely to rupture and bleed, indicating the coming wound remodeling and eventual stabilization of CSDH (**L**).

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
