# Peer review of "Subdural Lesions Linking Additional Intracranial Spaces and Chronic Subdural Hematomas: A Narrative Review with Mutual Correlation and Possible Mechanisms behind High Recurrence"

_diagnostics, 2023, doi:10.3390/diagnostics13020235_

Round 1

Reviewer 1 Report

The  Authors did a good job with the analysis and study design; the questions of this paper are relevant. The conclusions are sound and no additinal revision is necessary in order recommend publication.

Author Response

Point to point response as attached.

Reviewer 2 Report

Paper well written, concise enough. Clear description of the review scientific background, design and objectives.

Innovative interpretation of the disease with updated bibliographic references and very detailed images that better clarify what is described.

The review therefore fully achieves its educational information objective.

Author Response

Point to point response as attached.

Reviewer 3 Report

Although the intention of this review is to profoundly review and distinguish two different pathologies of the subdural space (Hygroma / chronic Subdural Hematoma), the review lacks of any structure, systematic literature search, and novelty. The abstract does not reflect the content of the manuscript, and the running title neither the abstract nor the manuscript itself.

Author Response

Point to point response as attached.
